# Cu-BTC Derived Mesoporous CuS Nanomaterial as Nanozyme for Colorimetric Detection of Glutathione

**DOI:** 10.3390/molecules29092117

**Published:** 2024-05-03

**Authors:** Xiwen Zhang, Jie Wang, Nan Chang, Yu Yang, Yuqi Li, Qi Wei, Chang Ni, Wanying Song, Mingyue Ma, Xun Feng, Ronghua Fan

**Affiliations:** 1School of Basic Medicine, Shenyang Medical College, Shenyang 110034, China; zxw13342493930@163.com; 2Department of Science and Technology, Shenyang Medical College, Shenyang 110034, China; wangjie0709@symc.edu.cn; 3Department of Sanitary Inspection, School of Public Health, Shenyang Medical College, Shenyang 110034, China; 17643315155@163.com (Y.L.); wq199715@163.com (Q.W.); 15942539626@139.com (C.N.); 13081263716@163.com (W.S.); 4Department of Food Science, School of Public Health, Shenyang Medical College, Shenyang 110034, China; changnan1984@126.com; 5Department of Physiology, School of Basic Medicine, Shenyang Medical College, Shenyang 110034, China; yangyu@symc.edu.cn; 6Department of Toxicology, School of Public Heath, Shenyang Medical College, Shenyang 110034, China; mymacmu@163.com; 7Department of Sanitary Chemisrty, School of Public Health, Shenyang Medical College, Shenyang 110034, China

**Keywords:** nanozyme, mesoporous CuS nanomaterial, Cu-BTC, glutathione, colorimetric detection

## Abstract

In this paper, Cu-BTC derived mesoporous CuS nanomaterial (m-CuS) was synthesized via a two-step process involving carbonization and sulfidation of Cu-BTC for colorimetric glutathione detection. The Cu-BTC was constructed by 1,3,5-benzenetri-carboxylic acid (H_3_BTC) and Cu^2+^ ions. The obtained m-CuS showed a large specific surface area (55.751 m^2^/g), pore volume (0.153 cm^3^/g), and pore diameter (15.380 nm). In addition, the synthesized m-CuS exhibited high peroxidase-like activity and could catalyze oxidation of the colorless substrate 3,3′,5,5′-tetramethylbenzidine to a blue product. Peroxidase-like activity mechanism studies using terephthalic acid as a fluorescent probe proved that m-CuS assists H_2_O_2_ decomposition to reactive oxygen species, which are responsible for TMB oxidation. However, the catalytic activity of m-CuS for the oxidation of TMB by H_2_O_2_ could be potently inhibited in the presence of glutathione. Based on this phenomenon, the colorimetric detection of glutathione was demonstrated with good selectivity and high sensitivity. The linear range was 1–20 μM and 20–300 μM with a detection limit of 0.1 μM. The m-CuS showing good stability and robust peroxidase catalytic activity was applied for the detection of glutathione in human urine samples.

## 1. Introduction

Glutathione (GSH) represents an essential antioxidant present widely in organisms, and plays an important role in a variety of metabolic processes. The abnormal change in its physiological level can directly contribute to cellular damage and many diseases, including liver disease, Alzheimer’s disease, HIV, and even cancer [1,2]. Thus, exploiting detection methods to trace GSH effectively and accurately is of great importance. Up to now, various analytical tools for the detection of GSH have been developed, such as chromatography [3], chemiluminescence [4], electrochemistry [5], molecular imprinting [6], fluorescent techniques [7], and colorimetric methods [8]. Among these methods, colorimetric assay is favored because of its advantages such as visual observation, simple operation, low cost, high efficiency, and fast response [9].

As a biocatalyst, natural enzymes, especially peroxidases have been widely used in colorimetric analysis. Natural peroxidases have the advantages of good substrate specificity and high catalytic efficiency, but they are also limited by the ease of denaturation, high sensitivity to environmental factors, time consuming in preparation, difficulty in purification, and special requirements for storage [10]. To overcome these shortcomings, there has been much work performed for the fabrication of enzyme mimics that can replace natural peroxidases. Since Fe_3_O_4_ nanoparticles were first reported to have intrinsic peroxidase activity [11], many different nanomaterials have been reported with such enzyme-like activity (known as nanozymes), including noble metal nanomaterials [12], carbon nanomaterials [13], transition metal oxides [14], transitional metal chalcogenides [15], and metal organic framework (MOF) [16]. Recently, nanozyme-based colorimetric systems have been also further developed for sensing in the fields of food detection [17], disease treatment [18], biomedical analysis [19] and environmental analysis [20], mainly because of their superiorities such as good stability, multifunctionalities, as well as low cost and efficient preparation.

Among the developed nanozymes, transitional metal dichalcogenides (TMDs) have attracted particular interest. For instance, Vipul et al. developed a Co_3_S_4_ nanosheet and successfully introduced it as a peroxidase mimic in a colorimetric sensor for the detection of L-cysteine [21]; Absalan et al. synthesized hierarchical hollow MoS_2_ nanotubes which also possessed peroxidase-like activity and could be used for colorimetric detection of D-penicillamine [22]. Especially among these TMDs, Cu-based TMDs have attracted a remarkable interest in nanozymes. Cu in its divalent form (Cu^2+^) could exhibit a similar effect to Fenton’s reagent and catalyze the conversion of hydrogen peroxide (H_2_O_2_) to hydroxyl (•OH) radicals, in which S could accelerate the charge transfer between H_2_O_2_ and Cu^2+^ [23].

Compared with solid counterparts, mesoporous nanostructures have attracted high interest because they greatly enlarge the specific surface area, exhibit excellent accessibility for internal space, enable sufficient mass and charge transport, as well as provide abundant active sites for catalytic reaction [24]. To date, some mesoporous nanomaterials—such as mesoporous MnFe_2_O_4_ magnetic nanoparticles [25] and iron-doped mesoporous silica nanoparticle [26]—have been reported with excellent nanozyme activity. Thus, there would be a great improvement in the enzyme-like activity of the CuS nanomaterials if they were designed with a mesoporous nanostructure. In recent years, nanomaterials derived from an organometallic framework offer an avenue and focus for the synthesis of porous nanomaterials. In this way, the inherent mesoporous properties of organometrics can be retained, while the stability and activity of the resulting nanomaterials can be also greatly improved [27].

In this study, an MOF-based synthetic strategy was utilized to prepare in situ m-CuS through successive carbonization and sulfidation. The synthesized m-CuS exhibited peroxidase-mimicking activity that enabled it to catalyze the oxidation of 3,3′,5,5′-tetramethylbenzidine (TMB) by H_2_O_2_. Furthermore, this property of the m-CuS nanozyme was introduced for the fabrication of a colorimetric sensor with the ability of selective detection of GSH, which showed a strong inhibitory effect on m-CuS nanozyme catalyzing TMB discoloration. Therefore, a sensitive colorimetric assay was exploited for the detection of GSH with fine linear ranges of 1–20 μM and 20–300 μM, as well as a detection limit as low as 0.1 μM. The m-CuS nanozymes with high catalytic activity explored in this work are expected to be applied for biosensing of other substances of interest.

## 2. Result and Discussion

### 2.1. Characterization of m-CuS Nanozyme

Figure 1 shows the morphology and microstructure of Cu-BTC and m-CuS obtained by SEM and HR-TEM. Before carbonization, the Cu-BTC precursor exhibited a typical octahedral morphology revealing a rough surface, a porous structure, and a size of about 200–500 nm, as shown in Figure 1a,b. After carbonization and sulfurization, this octahedron shape did not change much as shown in Figure 1c,d, although the surface turned rougher and the particle size increased marginally, which indicated that the in situ formed carbon originating from the Cu-BET could provide a rigid protection effect. The internal microstructure of the m-CuS was further characterized by HR-TEM, and the results are displayed in Figure 1e,f. From Figure 1e, it was obvious that m-CuS was totally incorporated in the octahedral carbon skeleton, which could be further elucidated by the larger version of the HR-TEM image in Figure 1f. The lattice fringes with the interspacing of 0.304 nm observed here could be attributed to the (102) crystal plane of CuS (Joint Committee on Powder Diffraction Standards (JCPDS) no. 79-2321) [28]. Moreover, local graphitization occurring at the edge carbon was also observed, which would be beneficial for fast electron transport due to the improvement of the electronic conductivity [29].

The XRD of Cu-BTC is shown in Figure 2a. The diffraction peaks at 6.83, 9.51, 11.70, 13.41, 15.11, 17.47, 19.13, 20.34, 24.11, 25.95, 29.33, 35.32 and 39.24° confirmed the cubic phase of the Cu-BTC, which was very consistent with that previously reported [30]. Figure 2b displays the XRD pattern of the synthesized m-CuS. The main diffraction peaks of m-CuS correspond to the standard CuS card (JCPDS no. 79-2321). The diffraction peaks at 2θ = 27.7, 29.3, 31.8, 39.9, 32.9, 47.9, 52.7, and 59.4°, are ascribed to (101), (102), (103), (006), (110), (108), and (116) planes of the m-CuS, respectively.

The EDX mapping of Cu-BTC is displayed in Appendix A (Appendix A). C, Cu, N, and O elements were observed, and they were of uniform distribution in the Cu-BTC sample. By contrast, the C, Cu, S, N and O elements (Appendix A) could be observed in m-CuS, which indicated the successful generation of CuS incorporated in the octahedral carbon framework. The elemental composition of m-CuS was further explained by XPS, as shown in Figure 3a–e. The chemical elements including Cu, S, C, and N, O could all be detected in the m-CuS, in which the O signal could have been caused by the exposure of the sample to oxygen in the air [28]. As shown in Figure 3b–e, the high-resolution Cu 2p, S 2p, C 1s, and N 1s XPS spectra were obtained. The two main peaks observed at 932.0 eV and 951.9 eV, respectively, in Figure 3b were identified as Cu 2p3/2 and Cu 2p1/2, while a weaker satellite peak observed at 945.6 eV could be caused by paramagnetic Cu^2+^ since it should be not observed for Cu (I) and Cu (0) [31]. According to Figure 3c, the peaks at 162.06, 163.2, and 163.9 eV could be ascribed to metal–sulfur bonding [32], and the peak at 168.8 eV might be attributed to some SOx species, which was probably caused by the adsorbed oxygen on the surface of m-CuS [33]. In addition, a side peak detected at 161.2 eV for m-CuS was in accordance with that found in a previous study [34]. In Figure 3d, the two peaks located at 284.6 eV and 286.9 eV, respectively, were ascribed to the sp^2^-hybridized graphitic C atoms (C-C/C=C) and the epoxy/alkoxyl groups (C-O). Moreover, the peak at 285.3 eV could be caused by the C-N bonds, demonstrating nitrogen doping in m-CuS [35]. In Figure 3e, the two peaks seen at 398.5 eV and 400.6 eV belonged to the pyridinic-N and the pyrrolic-N [36,37], respectively, further proving the evidence of N element in m-CuS.

The specific surface area and pore size distribution of Cu-BTC and m-CuS were detected by nitrogen adsorption/desorption analyses at 77.35 K. As shown in Appendix A (Appendix A), the Cu-BTC had a specific surface area of 210.857 m^2^/g, which was obtained based on the Brunauer–Emmett–Teller (BET) method. By virtue of the Barrett–Joyner–Halenda (BJH) method, the detailed information about the pore size distribution of Cu-BTC was obtained and is shown in Appendix A. The pore size in Cu-BTC was largely distributed at ~1.187 nm. After carbonization and sulfidation to m-CuS, the m-CuS had a specific surface area of 55.751 m^2^/g (Figure 4a). Moreover, the m-CuS displayed a type IV adsorption–desorption isotherm, in which a distinct hysteresis loop was found with high pressures (0.6–1.0 P/P_0_), revealing that the mesoporous nanostructure was present in the m-CuS [38]. The pore size in m-CuS largely distributed at ~15.38 nm (Figure 4b), which confirmed the mesoporous structure (<50 nm) of the m-CuS [38]. Such a structure with high porosity and specific surface would facilitate electrolyte infiltration and provide m-CuS with more abundant active sites [28].

### 2.2. Peroxidase-like Activity of m-CuS Nanozymes

To evaluate the peroxidase-like catalytic activity, the m-CuS catalyzed reaction of the peroxidase substrate TMB in the presence of H_2_O_2_ was investigated by comparison with the m-CuS-TMB and H_2_O_2_-TMB system. As evidenced by the inset of Figure 5a, the m-CuS was found to catalyze the TMB oxidation by H_2_O_2_ and produce a color change to the typical blue, which revealed that m-CuS possessed peroxidase-like catalytic activity.

Like HRP, the catalytic activity of m-CuS might be related to pH, temperature, and H_2_O_2_ concentration. Therefore, the peroxidase-like activity of m-CuS was further investigated by varying the temperature from 25 °C to 50 °C, pH from 1 to 10, and the H_2_O_2_ concentration from 0 to 300.0 mM, by comparison with HRP under experimental conditions with the parameters in the same range. The influence of these factors on the relative catalytic activity is shown in Figure 5b–d. In Figure 5b, the influence of temperature on the catalytic efficiency of m-CuS is displayed, where the optimal temperature was found to be 40 °C. A further increase in temperature would decrease the catalytic activity of m-CuS, probably because of the decomposition of H_2_O_2_. By contrast, the optimal temperature for the catalytic effect of HRP was 30 °C, which suggested that m-CuS was less sensitive to the reaction temperature. The effect of pH is shown in Figure 5c, and the greatest catalytic activities of m-CuS and HRP were both found at pH 4.0. The inset of Figure 5c indicates that when the pH was higher than 5.0, the solution did not produce a color change, but it produced an obvious blue color at pH 4.0. Additionally, it was observed that the obtained blue products became unstable with the increase of acidity, because they are subjected to further oxidization, which yielded yellow imide when the pH achieved 2.0 and 3.0. Based on these observation, pH 4.0 was selected for the following experiments. The effect of H_2_O_2_ concentration is shown in Figure 5d, which revealed that the maximal level of peroxidase activity of m-CuS was observed when the H_2_O_2_ concentration reached two times higher than HRP. However, when the H_2_O_2_ concentration was further increased, it rather decreased the peroxidase-like activity of the m-CuS. Therefore, the optimal concentration of H_2_O_2_ for m-CuS was obtained with 100.0 mM.

The apparent steady-state kinetic parameters were measured for the peroxidase-mimicking reaction, which were acquired by changing one substrate concentration while keeping the concentration of the other substrate constant. The results are shown in Figure 6a–d. Within a certain concentration range of the two substrates, the absorbance was divided by the molar absorption coefficient (39,000 M^−1^cm^−1^) of TMB-derived oxidation product to obtain a typical Michaelis–Menten curve for m-CuS. The related data were found to fit the Michaelis–Menten model. The Michealis–Menten constant (*K*_m_) was also calculated as an indicator of substrate affinity based on the Lineweaver–Burk plot:$$\frac{1}{v}=\left(\frac{K_{m}}{V_{max}}\right)\left(\frac{1}{\left[S\right]}\right)+\left(\frac{1}{V_{max}}\right)$$
where *v* represented the initial velocity, [*S*] represented the substrate concentration, *V*_max_ represented the maximal reaction velocity, and *K*_m_ represented the Michaelis constant. According to this calculation, the *K*_m_ and *V*_max_ of TMB were 0.827 mM and 5.075 × 10^−8^ Ms^−1^, and the *K*_m_ and *V*_max_ of H_2_O_2_ were 1.904 mM and 5.935 × 10^−8^ Ms^−1^. Furthermore, we compared the steady-state Michaelis–Menten kinetic constants obtained from m-CuS nanozymes with those from previously reported nanozymes. The result is shown in Table 1, which indicated that the *K*_m_ value of m-CuS when using H_2_O_2_ as the substrate was much lower than HRP [39], p-Co_3_O_4_ [40], MgFe_2_O_4_MNPs [41], Fe_3_O_4_/LNPs [42], and Zn-CuO [43], which suggested the superior affinity of m-CuS to H_2_O_2_. Simultaneously, the *K*_m_ value of m-CuS using TMB as the substrate was also lower than CuFe_2_O_4_ [44] and Zn-CuO [43], which was similar to that of HRP [39]. The greater TMB affinity of m-CuS might be due to the strong electrostatic attraction formed between the negatively-charged m-CuS and the positively-charged TMB substrate [2], which was further verified by the determined zeta potential values as shown in Appendix A (Appendix A). In addition, the *V*_max_ of m-CuS for TMB and H_2_O_2_ were greater than p-Co_3_O_4_ [40], CuFe_2_O_4_ [44], Fe_3_O_4_/LNPs [42], and Zn-CuO [43]. These results demonstrated that the as-prepared m-CuS could be employed as a potential alternative to natural HRP for detecting GSH.

### 2.3. The Catalytic Mechanism of m-CuS Nanozyme

For probing the possible mechanism for the peroxidase-like catalysis of m-CuS nanozymes, a fluorescence method was used in the following study. In brief, PTA was used to capture hydroxyl radicals generated by the m-CuS catalyzed H_2_O_2_ system. PTA itself has no fluorescence property, but it can easily combine with hydroxyl radicals and form 2-p-hydroxybenzoic acid, which has fluorescence property and can emit blue fluorescence at 430 nm. The result is shown in Figure 7a. When PTA was introduced to the m-CuS-H_2_O_2_ system, it was observed that the fluorescence intensity increased with time, which confirmed that m-CuS nanozyme could catalyze the decomposition of H_2_O_2_ to produce •OH radical and form 2-p-hydroxybenzoic acid with fluorescent properties with PTA. This experimental result confirmed that m-CuS nanozyme had similar peroxidase activity to other nanozymes [23].

### 2.4. Analytical Application for Determination of GSH

Figure 7b displays the UV-Vis absorption spectra of m-CuS-TMB-H_2_O_2_ detection system at varying concentrations of GSH. As shown in Figure 7b, when there was no GSH in the system, the absorbance value of m-CuS-TMB-H_2_O_2_ mixture was very high. When GSH was introduced to the above mixture, the absorbance value at 652 nm began to decline. For optimizing the experimental conditions for GSH sensing, the m-CuS-TMB-H_2_O_2_ system reaction time and GSH response time were explored. The result is shown in Figure 8a,b. A time-dependent absorbance of m-CuS-TMB-H_2_O_2_ system reaction was detected, which was first increased with the time from 5 min to 25 min, but then remained unchanged in the range 25 to 35 min with the greatest absorbance observed at 25 min as shown in Figure 8a. Furthermore, the effect of GSH response was explored, and the result is shown in Figure 8b. With the prolongation of the reaction time, the absorbance gradually decreased, and then achieved a platform at 5 min. Thus, the optimal reaction time and response time were determined to be 25 min and 5 min, respectively. Figure 7c shows the absorbance curves against the GSH concentration. As seen from Figure 7d–e, the absorbance value and GSH concentration present two linear relationships in the ranges of 1–20 μM and 20–300 μM, offering a detection limit of 0.1 μM. In Table 2, we compare different types of nanozymes for their analytical performance on the detection of GSH. The m-CuS exhibited improved performance, including a wider linear and the lowest detection limit, which were superior or at least comparable to most reported nanozyme sensors shown in Table 2.

### 2.5. Selectivity, Stability and Reusability

For testing the stability of the colorimetric detection platform for GSH determination, we conducted eight repeated measurements of GSH with the same concentration (50 μM) on one day, and the result is shown in Figure 9a. The experimental result showed that the catalytic system of m-CuS-TMB-H_2_O_2_ had good stability for GSH detection.

It was a challenge to determine GSH within complex human serum. To assess the anti-interference ability of the developed m-CuS based sensor, major coexistence of components, such as K^+^, Zn^2+^, glucose, glycine, lactose, and glutamate were taken into account. As shown in Figure 9b, there is no obvious effect on the determination of GSH for the above interferents. Hence, the designed m-CuS nanozymes based sensor was suitable for the detection of GSH.

### 2.6. Detection of GSH in Human Serum Samples

In order to confirm the practicability and feasibility of this proposed sensor for real samples, we detected the GHS in human serum. Recovery experiments were performed by addition of GHS in diluted human serum with final concentrations of 10, 30, 100, and 200 μM, respectively. The results are shown in Table 3, which indicates that the average recovery was in the range from 97.3% to 101.25% for all the samples, and the RSD was below 4.0%, verifying that this sensor could be reliable and applicable for the detection of GHS in real samples.

## 3. Materials and Methods

### 3.1. Reagents and Apparatus

Glutathione, 3,3′,5,5′-tetramethylbenzidine (TMB), and horseradish peroxidase (HRP) were obtained from Sigma (St. Louis, MO, USA). Poly-vinylpyrrolidone (PVP, K-30), 1,3,5-benzenetri-carboxylic acid (H_3_BTC), copper nitrate trihydrate (Cu(NO_3_)_2_·3H_2_O), terephthalic acid (PTA), sulfur, methanol, ethyl alcohol, hydrogen peroxide (H_2_O_2_, 30%), glucose, fructose, lactose, maltose, sodium monohydrogen phosphate (Na_2_HPO_4_), disodium hydrogen phosphate (Na_2_HPO_4_), sodium hydroxide (NaOH), and hydrochloric acid (HCl) were purchased from Sinopharm Chemical Reagent Co., Ltd. (Shanghai, China) All the chemicals were of analytical grade, and ultra-pure water was used in the study. Phosphate buffer solution (0.1 M) of different pHs was obtained by mixing the 0.1 M Na_2_HPO_4_ and 0.1 M NaH_2_PO_4_ in the water, after which the pH of the solution was adjusted either by HCl or NaOH. All the solutions were freshly prepared prior to each experiment.

UV-Vis absorption spectra were recorded on a UV-5200 UV-Vis spectrophotometer (Shanghai Xiwen Biotech. Co., Ltd., Shanghai, China). Fluorescence spectra were determined on an F-380 fluorescence spectrometer. Scanning electron microscopy (SEM) images were observed on a JSM-7800F scanning electron microscope (JEOL, Tokyo, Japan). Transmission electron microscope (TEM) images were recorded on a JEM-2100F transmission electron microscopy (JEOL, Japan). X-ray diffraction spectrometry (XRD) was measured on a smartlab 9 X-ray diffractometer (Rigaku, Tokyo, Japan) with a Cu-Kα radiation source (λ = 1.54056 Å). X-ray photoelectron spectroscopy (XPS) analysis was performed on an ESCALAB 250Xi X-ray photoelectron spectrometer (Thermo Fisher Scientific, Waltham, MA, USA) with an Al K alph source at a test energy of 1486.8 eV. The zeta-potential was measured using a NICOMP 380ZLS ζ Potential/Particle Sizer (PSS NICOMP, Santa Barbara, CA, USA). The specific surface area and pore size distribution of the samples were tested by ASAP-2020 Brunauer–Emmett–Teller (BET, Micromeritics, Norcross, GA, USA) based on the nitrogen adsorption–desorption isotherms.

### 3.2. Preparation of Cu-BTC

First, 3.6 g Cu(NO_3_)_2_·3H_2_O and 1.6 g PVP were fully dissolved in 200 mL methanol solution. BTC methanol solution was prepared by dissolving 1.72 g BTC in 200 mL methanol solution. Under the condition of constant stirring, the BTC methanol solution was then added to the mixture of Cu(NO_3_)_2_·3H_2_O and PVP, and the stirring was stopped immediately after the solution had been placed aside at room temperature for 24 h. The product was subsequently transferred to a centrifuge tube, centrifuged and triple-washed with methanol, and finally dried in vacuum at 60 °C to obtain blue Cu-BTC product for later use.

### 3.3. Synthesis of m-CuS

The m-CuS was synthesized according to a previously reported method [28]. Briefly, the Cu-BTC (0.4 g) was first carbonized under N_2_ for 2 h at 600 °C, with the heating rate of 2 °C min^−1^. The resulting black powder and sulfur powder were placed, respectively, on each end of the porcelain boat at a mass ratio of 1:2, where the sulfur powder was kept on the top and the black powder on the bottom. After that, the porcelain boat was transferred to a tube furnace, heated to 350 °C with a heating ramp of 2 °C min^−1^, and kept for 3 h under N_2_. m-CuS black powders were finally obtained after cooling to ambient temperature.

### 3.4. Enzyme Mimicking Properties of m-CuS Nanozyme

m-CuS could catalyze the conversion of TMB, a chromogenic substrate, into a colored product in the presence of H_2_O_2_. In the experiment, 20 μL TMB (10 mM), 20 μL m-CuS (1 mg/mL), 2.0 mL phosphate buffer solution (0.1 M, pH 4.0), and different amounts of H_2_O_2_ were mixed together in a 4.0 mL centrifuge tube. The sample was subsequently heated at 40 °C for 20 min in a water bath, and then measured by UV-Vis spectrophotometry.

Under the above experimental conditions, the catalytic reaction was carried out at different pH conditions (1.0–10.0) to test the influence of pH on the reaction. In a similar manner, the catalytic reaction was carried out at different temperatures (30–55 °C) to test the influence of temperature on the catalytic reaction. As a comparative experiment, the catalytic reaction of HPR was also carried out in accordance with the above experimental methods.

### 3.5. Kinetic Analysis

Based on the optimal experimental conditions, the steady-state kinetic investigations of m-CuS nanozymes were carried out using H_2_O_2_ and TMB as substrates. The m-CuS or HRP was mixed in 2.0 mL phosphate buffer solution with pH of 4.0 in the presence of H_2_O_2_ and TMB in a water bath at 40 °C. During the test, the concentration of TMB was changed without altering the concentration of H_2_O_2_, or the concentration of H_2_O_2_ was changed by keeping the concentration of TMB unchanged. The kinetic performance was obtained by recording the absorbance at 652 nm wavelength every 1 min for 5 min continuously under the kinetic test mode of the UV-Vis spectrophotometer.

### 3.6. The Investigation of m-CuS Nanozymes’ Peroxidase-like Catalytic Reaction Mechanism

For studying the catalytic mechanism of m-CuS nanozyme, the hydroxyl radical generated in the m-CuS-H_2_O_2_ system was detected by using PTA as a fluorescent probe. For specific experiments, 20 μL m-CuS (1 mg/mL), 20 μL H_2_O_2_ (30%), 1 mL PTA (20 mM), and 1960 μL phosphate buffer solution (pH 4.0) kept at a total volume of 3.0 mL were mixed together at room temperature. The fluorescence intensity at different reaction times was then determined by a fluorescence spectrophotometer.

### 3.7. Detection of Glutathione

The detection of GSH was achieved through the following experimental methods. First, 1850 μL phosphate buffer solution (0.1 M, pH 4.0), 20 μL TMB (10 mM), 10 μL H_2_O_2_ (1.0 M), and 20 μL m-CuS (1.0 mg/mL) were mixed. After that, the solution was heated at 40 °C for 20 min, and then 100 μL GSH solution with different concentrations was added. The solution was kept at 40 °C for an additional 5 min. The absorbance value was recorded at 652 nm by a UV-Vis spectrophotometer.

## 4. Conclusions

A novel colorimetric sensor based on m-CuS nanozymes was developed for the quantitative detection of GSH, and was prepared via carbonization and sulfidation of Cu-BTC. It was found that the m-CuS had excellent peroxidase-like activity, which could efficiently catalyze the oxidation of the substrate TMB in the presence of H_2_O_2_. Benefiting from this activity of m-CuS nanozymes, the quantitative analysis of GSH could be realized via the competing reactions of GSH with the chromogenic substrates led by the •OH radical, which demonstrated a wide linear range form 1–20 μM and 20–300 μM, a low detection limit of 0.1 μM, and good recoveries. At the same time, some limitations of this work also exist, such as the catalytic activity and selectivity of m-CuS which should be further improved. In summary, this work provided a new method for the synthesis of mesoporous CuS nanozymes, and more significantly, the developed m-CuS nanozymes showed a great potential for use in other biosensors and in biocatalysis.

## Figures and Tables

**Figure 1 molecules-29-02117-f001:**
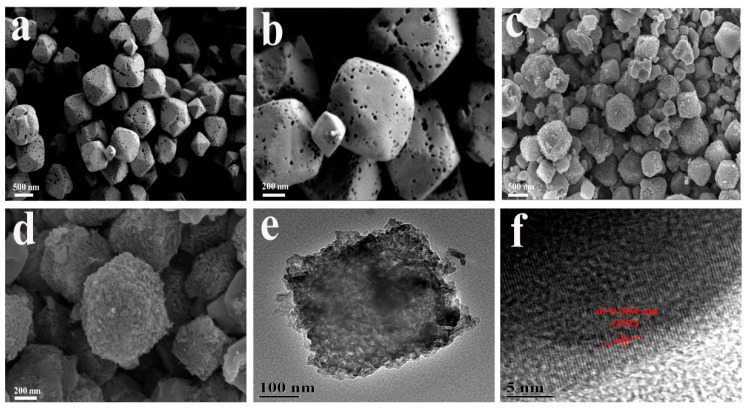
SEM images (**a**,**b**) of the Cu-BTC precursor; SEM (**c**,**d**), HRTEM (**e**,**f**) images of the m-CuS.

**Figure 2 molecules-29-02117-f002:**
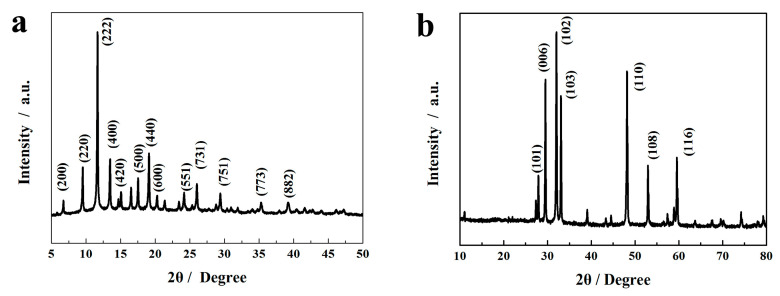
XRD pattern of Cu-BTC (**a**) and m-CuS (**b**).

**Figure 3 molecules-29-02117-f003:**
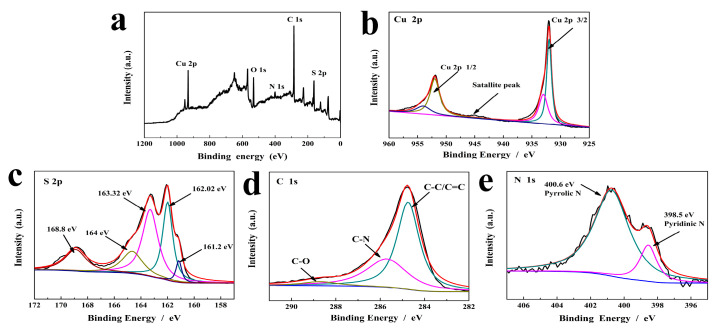
(**a**) Survey scan XPS survey spectrum of m-CuS; and (**b**–**e**) high resolution XPS spectra of the Cu 2p, S 2p, C 1s, and N 1s levels in m-CuS.

**Figure 4 molecules-29-02117-f004:**
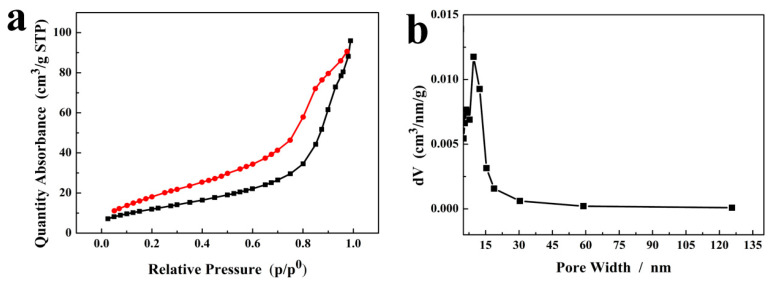
(**a**) Nitrogen adsorption–desorption isotherm; (**b**) Pore size distribution of the m-CuS.

**Figure 5 molecules-29-02117-f005:**
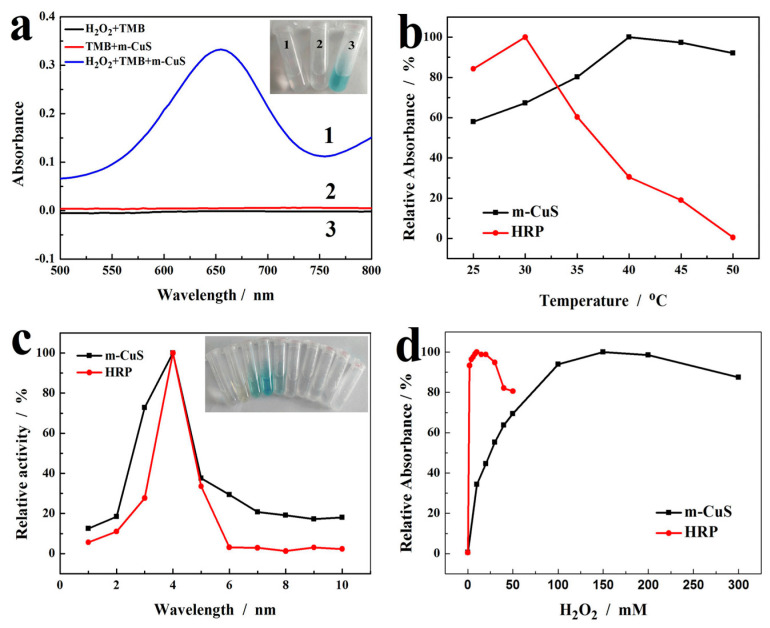
(**a**) UV-Vis spectra of (1) TMB-H_2_O_2_, (2) m-CuS-TMB, and (3) m-CuS-TMB- H_2_O_2_. (The inset shows a picture of the above solutions). The catalytic performance of m-CuS and HRP on TMB under different temperature (**b**), pH (**c**) (the inset of (**c**) shows the photograph of oxidized TMB by m-CuS in aqueous solutions with different pH values), and H_2_O_2_ concentration (**d**).

**Figure 6 molecules-29-02117-f006:**
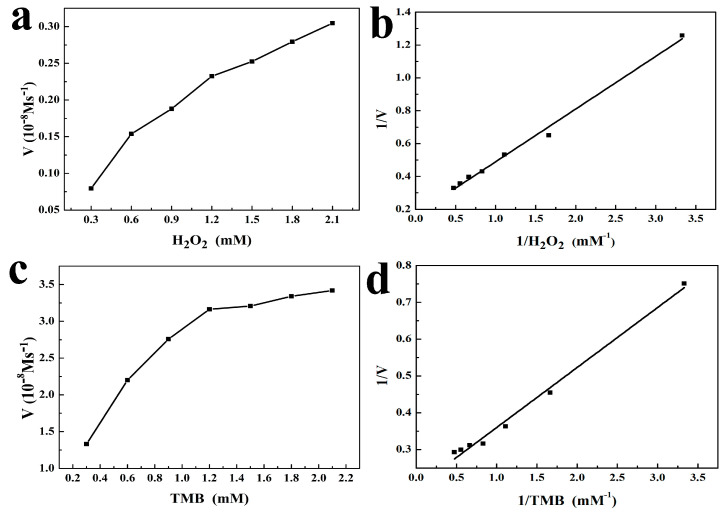
Steady-state kinetic analysis of m-CuS as nanozyme. (**a**) The TMB concentration was 500 μM and the concentration of H_2_O_2_ was varied. (**b**) Lineweaver–Burk double reciprocal plots of (**a**). (**c**) The H_2_O_2_ concentration was 30 mM and the concentration of TMB was varied. (**d**) Lineweaver–Burk double reciprocal plots of (**c**).

**Figure 7 molecules-29-02117-f007:**
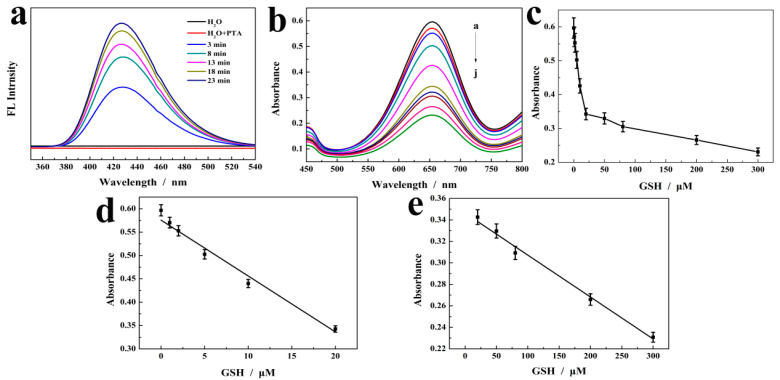
(**a**) The fluorescence intensity of m-CuS-H_2_O_2_ system generated •OH radicals captured by PTA changes with time. (**b**) UV-Vis absorption spectra of m-CuS-TMB-H_2_O_2_ with different concentrations of GSH; (**c**) the change in absorbance at 652 nm at different concentrations of GSH; the linear calibration plots of GSH (**d**) 1 to 20 μM and (**e**) 20 to 300 μM.

**Figure 8 molecules-29-02117-f008:**
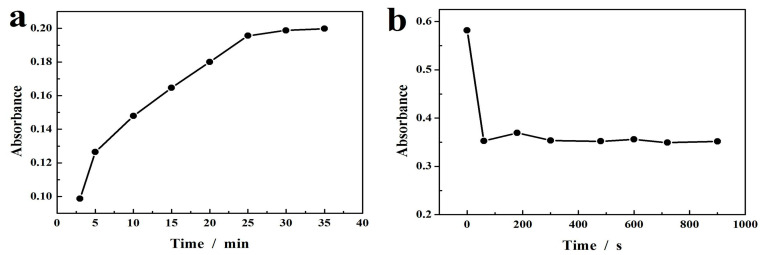
(**a**) Effect of reaction time on the m-CuS-TMB-H_2_O_2_ system. (**b**) Effect of GSH response time on the m-CuS-TMB-H_2_O_2_ system for the GSH detection.

**Figure 9 molecules-29-02117-f009:**
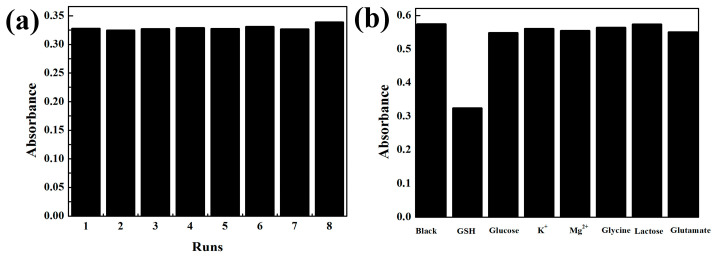
(**a**) Repeatability of m-CuS-TMB-H_2_O_2_ for 8 runs determination of GSH (50 μM). (**b**) The selectivity of m-CuS-TMB-H_2_O_2_ for GSH (100 μM) detection; interference substances were glucose (1000 μM), K^+^ (1000 μM), Mg^2+^ (1000 μM), glycine (1000 μM), lactose (1000 μM), and glutamate (1000 μM).

**Table 1 molecules-29-02117-t001:** Comparison of Michaelis–Menten constants of m-CuS with the other previously reported nanozymes.

	*K*_m_ (mM)		*V*_m_ (10^−8^ M s^−1^)		
Catalyst	H_2_O_2_	TMB	H_2_O_2_	TMB	Ref.
HRP	3.70	0.43	8.71	10.00	[39]
p-Co_3_O_4_	3.43	0.12	1.03	1.14	[40]
CuFe_2_O_4_	0.50	2.26	2.61	2.07	[44]
MgFe_2_O_4_MNPs	4.61	0.67	13.46	2.09	[41]
Fe_3_O_4_/LNPs	5.30	0.51	0.96	1.03	[42]
Zn-CuO	7.10	10.00	0.30	2.88	[43]
m-CuS	1.90	0.83	5.94	5.08	This work

**Table 2 molecules-29-02117-t002:** Performances comparison of m-CuS with the other previously reported nanozymes in determination of GSH.

Sensing Probe	Linear Range (μM)	LOD (μM)	Reference
MnO_2_ CD	1–10	0.3	[45]
BSA-Au NCs	10–400	0.12	[46]
MnO_2_ NS-TMB	1–25	0.3	[47]
MnO_2_-GQD	0.5–10	0.15	[48]
PEI-Ag NPs	0.5–6	0.38	[49]
Cu-CuFe_2_O_4_	2.5–10	0.31	[44]
Au nanoclusters	2–25	0.42	[50]
D-ZIF-67	0.5–10	0.2292	[51]
Sb-FeOCl	1–36	0.495	[52]
AuNPs-MIP	5–40	1.16	[53]
Hemin/GQD	1–50	0.2	[54]
m-CuS	1–20; 20–300	0.1	This work

CD: carbon dots; NC: nanocrystal; NP: nanoparticle; PEI: polyethyleneimine; MIP: molecularly imprinted polymer.

**Table 3 molecules-29-02117-t003:** Determination of GSH in human serum samples using m-CuS nanozymes.

Sample	GSH Added (μM)	GSH Found (μM)	Recovery (%)	RSD (%, n = 3)
1	10	9.8	98.0	2.4
2	30	29.2	97.3	3.5
3	100	99.6	99.6	3.4
4	200	202.5	101.25	2.6

## Data Availability

Data are contained within the article and Appendix A.

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
