# Peer review of "Cu-BTC Derived Mesoporous CuS Nanomaterial as Nanozyme for Colorimetric Detection of Glutathione"

_molecules, 2024, doi:10.3390/molecules29092117_

Round 1
Reviewer 1 Report
Comments and Suggestions for Authors
Requires revision.

Reviewer 2 Report
Comments and Suggestions for Authors
The authors of this manuscript prepared mesoporous catalyst materials using a two-step approach and gave satisfactory characterization (XRD, XPS, SEM, HR-TEM, BET). It was applied in the oxidation of 3,3',5,5'-tetramethylbenzidine. In additional studies they performed mechanistic studies with the conclusion that active oxygen species are involved in the oxidation process.
Three minor remarks
i) Title: …CuS Nanomaterials… Why plural? There is only a single catalyst?
ii) according to Fig. 8a catalytic activity in a reuse test increased in run 8. Any reason?
iii) “Time” is indicated – it should be runs.
Development of a new catalyst and explore its peroxidase-like activity. Development of a new catalyst with results improved in comparison to previous findings. The catalyst works as a colorimetric sensor affording quantitative detection of GSH. None – results in comparison to previous studies clearly show the advantage of their preparation.
i) development of a new, mesoporous catalyst –yes ii) colorimetric detection methods to detect GSH –yes iii) peroxidase-like activity mechanism studies –yes iv) kinetic analysis –yes
References are appropriate.
All figures (1–8) are of high quality. All tables (1–3) are clearly organized and data are arranged as required. Tables 1 and 2 provide readers a possibility to compare results with those reported in previous studies.
